# Switchable Optical Properties of Dyes and Nanoparticles in Electrowetting Devices

**DOI:** 10.3390/nano14020142

**Published:** 2024-01-09

**Authors:** Urice N. Tohgha, Jack T. Ly, Kyung Min Lee, Zachary M. Marsh, Alexander M. Watson, Tod A. Grusenmeyer, Nicholas P. Godman, Michael E. McConney

**Affiliations:** 1Air Force Research Laboratory, Materials and Manufacturing Directorate, Wright-Patterson AFB, OH 45433, USA; urice.tohgha.ctr@us.af.mil (U.N.T.); kyungmin.lee.3.ctr@us.af.mil (K.M.L.); zachary.marsh.2.ctr@us.af.mil (Z.M.M.); tod.grusenmeyer.1@us.af.mil (T.A.G.); nicholas.godman.2@us.af.mil (N.P.G.); 2Azimuth Corporation, Fairborn, OH 45431, USA; 3UES, Inc., Dayton, OH 45432, USA; jack.ly.ctr@us.af.mil; 4Department of Engineering Management, School of Engineering, Systems, and Technology, University of Dayton, Dayton, OH 45469, USA

**Keywords:** electrowetting, device pixels, electrical switching, BODIPY dyes, nanoparticles, transmittance spectra

## Abstract

The optical properties of light-absorbing materials in optical shutter devices are critical to the use of such platforms for optical applications. We demonstrate switchable optical properties of dyes and nanoparticles in liquid-based electrowetting-on-dielectric (EWOD) devices. Our work uses narrow-band-absorbing dyes and nanoparticles, which are appealing for spectral-filtering applications targeting specific wavelengths while maintaining device transparency at other wavelengths. Low-voltage actuation of boron dipyromethene (BODIPY) dyes and nanoparticles (Ag and CdSe) was demonstrated without degradation of the light-absorbing materials. Three BODIPY dyes were used, namely Abs 503 nm, 535 nm and 560 nm for dye 1 (BODIPY-core), 2 (I_2_BODIPY) and 3 (BODIPY-TMS), respectively. Reversible and low-voltage (≤20 V) switching of dye optical properties was observed as a function of device pixel dimensions (300 × 900, 200 × 600 and 150 × 450 µm). Low-voltage and reversible switching was also demonstrated for plasmonic and semiconductor nanoparticles, such as CdSe nanotetrapods (abs 508 nm), CdSe nanoplatelets (Abs 461 and 432 nm) and Ag nanoparticles (Abs 430 nm). Nanoparticle-based devices showed minimal hysteresis as well as faster relaxation times. The study presented can thus be extended to a variety of nanomaterials and dyes having the desired optical properties.

## 1. Introduction

The utilization of liquid-based optical shutter devices for virtual reality applications is important from both an industrial and academic perspective. Devices based on electrowetting-on-dielectric (EWOD) are particularly appealing due to the rapid actuation speeds [1,2,3], lower power consumption [4] and the flexibility in the design of the substrates (soft as well as rigid substrates with relatively easy fabrication methods) [5,6]. Electrowetting-on-dielectric (EWOD) represents an electrowetting setup comprising a polar droplet/fluid on an insulating thin-film dielectric material coated on a conductive substrate, whereby the wetting properties of the droplet/fluid can be controlled by applying a voltage between the conductive droplet/fluid and underlying conductive substrate [7,8,9,10,11,12]. The EWOD platform also enables practical applications such as visible light and IR light shutters [13,14,15]. The unique device architecture of EWOD devices permits the incorporation of a wide range of light-absorbing materials having optical properties spanning the UV to IR region [14,16]. While Ren and coworkers showed that a neat glycerol droplet could be used as an IR light-absorbing material [14,17], dyes and inks are mostly used as the light-absorbing materials in these devices [18]. Ribet et al. [13] used a water-soluble dye, patent blue VF, to achieve maximum light attenuation with 20% dye concentration in ionic liquids. The device had two pairs of electrodes on a substrate and a hole in the device. Although this device had the advantage of low insertion loss, reliable optical attenuation could only be achieved through very small holes in the device. Lee et al. used a black ink solution in an optical shutter device for the security of electronic devices [19,20]. This device had a 5 mm circular hole as the active area of the device. The authors utilized sequential actuation of electrodes to move droplets from one electrode to the other. It is also worth noting that the aforementioned studies and related literature lay more emphasis on the device architecture, assembly and switching speeds (typically milliseconds for electrowetting devices) [21,22,23,24,25,26,27] while there are very limited studies centered on changing the optical properties of the light-absorbing materials in the device. Materials having robust and tunable optical properties are desired for more practical applications.

We present work on low-voltage actuation of electrowetting (EW) devices dosed with both dyes and nanoparticles (NPs). Our work focuses on the changes in the optical properties of the light-absorbing materials as a function of the applied field. Unlike previous studies focusing on broadband-absorbing black inks/dyes [13,22], our work uses custom-synthesized narrow-band-absorbing dyes and nanoparticles. Such optical properties are appealing for spectral filtering devices targeting specific wavelengths while maintaining device transparency at other wavelengths. Boron dipyromethene (BODIPY) dyes with high extinction coefficients and narrow absorption bands were first explored. Three BODIPY dyes (structures shown in Figure 1) were used and will be referred to as dyes 1, 2 and 3 corresponding to BODIPY-core, I_2_BODIPY and BODIPY-TMS, respectively. The dyes were explored in devices having varying pixel dimensions with the aim of realizing reversible actuation. All three dyes were successfully actuated in 300 × 900 devices, but reversible actuation was not achieved; however, reversible actuation was observed in 200 × 600 devices (using dye 3 as a representative dye). The results were mainly attributed to the surface tension of dye 3 (higher than the surface tensions of dyes 1 and 2) as well as the device pixel dimensions. This suggests that the dye structure is also an important factor in optical shutter devices.

Our work also employed nanomaterials as light-absorbing materials in devices. Nanoparticles with relatively narrow absorption spectra were chosen so that devices having discernible spectra and peaks could be assembled. Plasmonic Ag nanoparticles (12 nm average size), semiconductor CdSe nanotetrapods (2 nm average arm width and 27 nm average arm length) and CdSe nanoplatelets (average length 6 nm, average width 2 nm, thickness 1.4 nm) [28] were used as representative nanomaterials in electrowetting devices. The absorption spectra are shown in the ESI. Low-voltage and reversible switching of the optical properties for all NPs was achieved. These studies can thus be extended to a diverse range of nanomaterials having different optical properties.

## 2. Materials and Methods

Materials: Mesitaldehyde (98%, Aldrich, Milwaukee, WI, USA), 2,4-Dimethylpyrrole (>97%, TCI America, Portland, OR, USA), Trifluoroacetic acid (99%, Aldrich), p-Chloroanil (>95%, Aldrich), Boron trifluoride diethyl etherate (>98%, TCI America), Triethylamine (99%, Alfa Aesar, Ward Hill, MA, USA), N-idosuccinimide (95%, Aldrich), Sodium thiosulfate (99%, Aldrich), Sodium sulfate (Aldrich), trans-Dichlorobis(triphenylphosphine) palladium (II) (99%, Strem Chemicals, Newburyport, MA, USA), Copper (I) iodide (98%, Aldrich), Sodium Chloride (99%, Oakwood Products, Estill, SC, USA), Trimethylsilylacetylene (98%, Oakwood Products), Silver nitrate (AgNO_3_, 99.999%, Aldrich), Oleic acid (OA, 90%, Aldrich), Oleylamine (OLA, 70%, Aldrich), Octadecene (ODE, 90%, Aldrich), Cadmium oxide (CdO, 99.99%, Aldrich), Cetyl trimethylammonium bromide (CTAB, 98%, Alfa Aesar), Trioctylphosphine (TOP, 90%, Strem chemicals), Cadmium acetate (Cd(Ac)_2_, Aldrich), Selenium powder (Se, 99.9%, Alfa Aesar), Dichloromethane (Fisher Chemical, Waltham, MA, USA), n-Hexanes (Fisher Chemical), Ethyl Acetate (Fisher Chemical), tetradecane (Aldrich) and limonene (Aldrich) were used as received.

Dyes and nanoparticle synthesis: Ag NPs were synthesized following the procedure reported by Li et al. [29]. Briefly, AgNO_3_ (170 mg, 1 mmol), OA (10 mL) and OLA (10 mL) were added to a three-neck flask and heated to 190 °C for 1 h. The product was purified by precipitation with ethanol and precipitate dispersed in toluene. The average nanoparticle size was 13 nm.

CdSe nanotetrapods were synthesized following the literature procedure reported by Wong and coworkers [30]. Briefly, ODE (20 mL), OA (2 mL, 6.28 mmol) and CdO (128.4 mg, 1 mmol) were added to a three-neck flask. The mixture was degassed and heated to 200 °C under argon flow to form Cd(oleate)_2_. A separate mixture of the selenium precursor was prepared by dissolving Se powder (39.5 mg, 0.5 mmol) in 1.5 mL TOP and adding CTAB suspension (0.02 g in 3 mL toluene)) to the degassed mixture. The TOPSe/CTAB mixture was quickly injected into the cadmium precursor at 190 °C and the reaction was allowed to run for 3 min before being stopped. Purification was carried out by repeated precipitation with ethanol and dissolution in hexanes. Nanoparticle size and size distribution were: average arm length = 27.3 ± 1.6 nm and average arm width = 2.1 ± 0.2 nm.

CdSe nanoplatelets were synthesized following modified literature methods [28,31]. Briefly, OA (0.3 mL), ODE (15 mL) and Cd(Ac)_2_ (240 mg) were loaded in a 100 mL three-neck flask and degassed at 80C for 1 h. The reaction flask was heated to 200 °C and kept under nitrogen flow. Selenium precursor was prepared separately by dissolving Se (180 mg) in TOP (2 mL) under nitrogen. Selenium precursor was injected into the Cd precursor and the reaction was allowed to run for 30 min. before being quenched in a water bath. Purification was carried out by precipitating once with butanol/methanol mixture and precipitates were dissolved in hexanes prior to use. BODIPY-core (1) was synthesized as previously reported [32]. Nanoparticle size and size distribution were: average arm length = 6.9 ± 1.9 nm and average arm width = 2.4 ± 0.4 nm.

Precursor 1,3,5,7-Tetramethyl-8-mesityl-4,4-difluoroboradiazaindacene was synthesized by reacting mesitaldehyde (0.677 g, 4.57 mmol), 2,4-dimethylpyrrole (1 g, 10.51 mmol) and 2 drops of trifluoroaceitc acid in 20 mL anhydrous dichloromethane under inert conditions for 3 h at room temperature. Then, p-chloranil (1.12 g, 4.57 mmol) was quickly added and stirred for 1 h. The solution was then cooled in an ice bath where excess triethylamine (2.64 g, 3.36 mL, 26 mmol) and excess boron trifluoride diethyl etherate (5.19 g, 4.51 mL, 36.6 mmol) were sequentially added dropwise via syringe. The reaction was warmed to room temperature and stirred for 1 h. The precipitate formed in the reaction was isolated via vacuum filtration. The filtered solid was washed with dichloromethane. The filtrate was dried and purified via silica chromatography with a hexane/ethyl acetate 10:1 (*v*:*v*) mixture as the eluent. The product fraction was collected and the solvent was removed via rotary evaporation yielding orange crystals. Yield: 1.14 g (68%). ^1^H NMR (400 MHz, CDCl_3_): 6.94 (s, 2H), 5.96 (s, 2H), 2.56 (s, 6H), 2.33 (s, 3H), 2.09 (s, 6H), 1.38 (s, 6H). MW: 366.18 g/mol.

I_2_BODIPY (2) was synthesized by dissolving 1,3,5,7-Tetramethyl-8-mesityl-4,4-difluoroboradiazaindacene (100 mg, 273 µmol) in 10 mL of DCM in a 100 mL two-neck round-bottom flask. Recrystallized N-iodosuccinimide (135 mg, 601 µmol) was added and stirred in the dark at room temperature for 12 h. The DCM solution was then washed with 10% sodium thiosulfate aqueous solution. The organic solution was collected and dried over sodium sulfate. The solvent was then removed via rotary evaporation. The crude product was purified via silica chromatography using a hexane/ethyl acetate (9:1 *v*:*v*) mixture as the eluent. The product fraction was collected and the solvent was removed via rotary evaporation yielding orange-like, green iridescent crystals. Yield: 110 mg (65%). ^1^H NMR (400 MHz, CDCl_3_) δ 6.97 (s, 2H), 2.65 (s, 6H), 2.36 (s, 3H), 2.06 (s, 6H), 1.40 (s, 6H). LDI-MS: PhMe_3_-I_2_BDP calculated mass: 615.996437. Found mass: 615.99535 (1.8 ppm error).

Precursor 2,6-diiodo-1,3,5,7-tetramethyl-8-phenyl-4,4-difluoroboradiazasindacene) was synthesized as previously reported [32].

BODIPY-TMS (3) was synthesized using a modification of a previously reported synthetic procedure [33]. BODIPY-TMS (3) was synthesized by dissolving 2,6-diiodo-1,3,5,7-tetramethyl-8-phenyl-4,4-difluoroboradiazasindacene (2 g, 3.5 mmol) Pd(PPh_3_)_2_Cl_2_ (243 mg, 0.35 mmol) and CuI (75 mg, 0.417 mmol) in a 3:1 THF:TEA mixture under inert atmosphere. Trimethylsilylacetylene (1.02 g, 10.4 mmol) was added dropwise via syringe. The reaction was stirred for 5 h at 55·°C. After cooling to room temperature, the solution was washed with brine and dried over sodium sulfate. The solvent was then removed via rotary evaporation. The crude product was purified via silica chromatography using a hexane/ethyl acetate 9:1 (*v*:*v*) mixture as the eluent. The product fraction was collected and the solvent was removed via rotary evaporation yielding orange/red crystals. Yield: 1.48 g (83%). ^1^H NMR (400 MHz, CDCl_3_): 7.49–7.47 (m, 3H), 7.29–7.26 (m, 2H), 2.64 (s, 6H), 1.45 (s, 6H), 0.20 (s, 18H).

Characterization: The absorption spectra of dye and NPs solutions were collected via a Cary UV-vis-NIR spectrophotometer. Transmission electron microscopy (TEM) images were collected using a Talos 200 KV transmission electron microscope (Thermo Fisher Scientific, Waltham, MA, USA). Surface tension measurements of fluids were performed using the pendant drop technique with a computer-controlled Attension Theta Optical Tensiometer instrument. Surface tension values were obtained directly from the analysis using the instrument software. All devices were characterized by measuring transmittance spectra with and without applied field using a function generator and voltage amplifier (AC field 0–20 V, 400 Hz). The waveform generator was connected to the voltage amplifier using BNC cables, and the voltage amplifier was connected to cables with alligator clips. These were attached to the leads on the device before applying voltage. Transmission spectra were collected with a fiber optic spectrometer (Ocean Optics).

Electrowetting Device assembly and operation: Pixelated devices were obtained from Adroit R&D LLC Company and were assembled using the techniques recommended by the company [22]. Briefly, the pixelated device substrate was attached to a linear actuator pending submersion into the biphasic fluid system (water background and oil/dye/nanoparticle). The device’s top substrate (ITO-coated glass), binder clips and gasket were immersed in the water bath. A fluid isolation tube (to contain the oil) was attached to the water bath and the oil mixture was added (40–100 µL) at the center of the tube. The linear actuator was turned ON and the pixelated substrate was lowered into the water bath by first going through the oil in the isolation tube. The oil/dye/nanoparticle fluid was trapped in the pixels by the water in the pixel grids. The two substrates were assembled using a gasket and binder clips. The assembled device was dried with air and kept for characterization. Figure 1 below depicts the actuation of a representative device using a black fluid (to show images with better contrast in the ON and OFF states). When no voltage is applied (V = OFF), the oil/dye mixture is trapped in the device pixels and the water is in the pixel grid. The application of voltage results in actuation-induced oil/dye dewetting from the device pixels and the pixels are ‘filled’ with water. There is thus greater transparency of the device pixels (V = ON, Figure 1). The devices presented in this work all operate in a similar fashion. The active area of the devices is 2.5 cm × 2.5 cm.

## 3. Results

### 3.1. Dye Absorption Spectra, Structures and Extinction Coefficients

The dye structures and absorption spectra are shown in Figure 1. Three dyes having identical core structures but with distinct optical properties were chosen. The extinction coefficients of the dyes (shown in Figure 1) are indicative of highly absorptive dyes, which are suitable for absorptive applications. The high absorptivity also ensures that relatively low dye concentrations (8 mg/mL) can be used. The choice of dyes in such devices is considered the most challenging aspect [22] and our custom-synthesized BODIPY dyes are well-suited for exploring the switchable optical properties in devices. The three dyes were prepared in a limonene/tetradecane (4 to 1 *v*:*v*) solvent system. Limonene is critical to the solubility of the dye (the dyes are more soluble in limonene than tetradecane) while tetradecane was used to modulate the surface tension of the fluids to desired values. The solvent system and dye concentration (8 mg/mL) were chosen to maximize the solubility of all three dyes and to achieve identical dye concentrations in devices.

### 3.2. Actuation of Dyes in 300 × 900 µm (Pixel Dimensions) Devices

Devices having 300 × 900 µm pixel dimensions (devices with the widest pixel dimensions) were tested first. A blank control test was performed and the spectrum is shown in Appendix A. Figure 2 presents transmittance spectra for the electro-optic response of the different dyes in devices. Low-voltage actuation (7.5 V) was achieved for all three dyes. The spectra in Figure 2a show a 32.6% reduction in the peak intensity of dye 1 (at 503 nm) at the initial application of voltage (7.5 V). Further increasing the voltage (10 V) resulted in a more gradual change in peak intensity (10%). Figure 2b shows similar results for dye 2 where a 33.1% reduction in peak intensity (at 535 nm) is observed with the application of voltage (7.5 V). The peak intensity for dye 2 is further reduced by 10% at 10 V. Interestingly, dye 3 showed the greatest change in peak intensity (43.1%) at 7.5 V and a further 9.8% peak intensity reduction at 10 V. There was thus >50% overall decrease in the peak intensity of dye 3, unlike with dyes 1 and 2. The results suggest that the dye structure may influence the electro-optic performance of the devices. This is likely due to the subtle differences in the compatibility of the different dyes with the solvent system as a result of different substituents on the core structure. The reversible actuation of devices is critical for any type of practical application. The effect of hysteresis on the performance of all three devices was evaluated by taking spectra before and after voltage application as shown in Appendix A for dyes 1, 2 and 3, respectively. All three devices showed pronounced hysteresis. The peak intensity recovered for each dye after the removal of voltage was 68%, 70% and 76% for dyes 1, 2 and 3, respectively. Interestingly, the device with dye 3 showed the least hysteresis, which suggests that dye 3 is more compatible with the device/solvent system than the other two dyes. The hysteresis in such devices is typically driven primarily by the surface tension of the fluids and the device pixel dimensions used [22]. Zhou et al. showed that the interfacial tension forces cause the dye/solvent system to recover the pixel surface when the voltage is turned off [22]. The surface tension of the dyes used in this work was measured and is shown in Section 3.3.

### 3.3. Surface Tension of Dyes 1, 2 and 3

The response of the device to voltage is driven by the interfacial surface tension between the fluids [34]. The surface tensions of all three dye solutions were measured to better understand the wettability properties. Pendant drop measurements were carried out in air. The dyes were dispersed in limonene, which has a lower density than water, and hence, pendant drop measurements in the biphasic system will result in droplets forming at the surface of the water. Table 1 shows the values for all three dyes at an identical concentration (8 mg/mL). Low surface tension in fluids has been shown to cause significant contact angle hysteresis when voltage is removed [22]. This corroborates the more pronounced hysteresis effects observed for dyes 1 and 2 compared to dye 3. Hence, dye 3 was chosen for further studies using devices of different pixel dimensions.

### 3.4. Actuation of Dye 3 in Devices with 200 × 600 and 150 × 450 µm Pixel Dimensions

The transmittance spectra corresponding to the initial actuation of dye 3 in a device with 200 × 600 pixel dimensions are shown in Appendix A. An image of the corresponding device is shown in Figure 3b. Interestingly, there was no actuation at 0–10 V (no change in dye peak intensity), which suggests that pixel dimensions influence the actuation voltage, as previously observed by Cheng and coworkers [23]. The authors reported that the operating voltage increased with decreasing pixel dimensions (width/length). This was attributed to the need to have a smaller water contact angle with the pixel substrate, which is achieved with increased voltage. A further increase in voltage (15 V) resulted in device actuation and a 79% decrease in dye peak intensity (Figure 3a, red curve). More importantly, the 200 × 600 pixel device showed complete recovery of dye peak intensity when the voltage was removed (Figure 3a, dashed green curve). The results strongly indicate that both the surface tension and device pixel dimensions are critical to realizing switchable optical properties of dyes at low voltages.

Repeated actuation of the device (three cycles) showed no significant hysteresis effects (Appendix A), which is desired for practical applications. To further explore the effect of pixel dimensions on device performance, a third device having 150 × 450 µm pixel dimensions was used. The same concentration of dye 3 was dosed in the device and the transmittance spectra are shown in Appendix A. No actuation of the device was achieved (no changes in dye peak intensity) even at higher voltages (30 V). While the application of higher voltages could potentially result in the actuation of the device, dielectric breakdown (leading to irreversible device performance) is likely to occur at such high voltages. Moreover, this work was focused on low-voltage actuation (0–20 V) of light-absorbing materials in devices.

### 3.5. Devices Dosed with Nanoparticles

The exploration of light-attenuating materials in liquid-based optical shutter devices was extended to inorganic NPs having relatively narrow absorption spectra (Appendix A). While dyes have been extensively explored in most electrowetting and optical shutter devices, dyes are susceptible to photo-degradation compared to inorganic materials [16,21]. In this work, we first explored silver NPs as representative plasmonic NPs.

#### 3.5.1. Ag Nanoparticles in EW Device

Ag NPs (Abs 430 nm) dispersed in a limonene/tetradecane (4 to 1 *v*: *v*) solvent system were dosed in the device in a similar fashion to the dyes reported above. The concentration of NPs (30 mg/mL) was chosen to minimize NP aggregation and maximize reversible device actuation, which is dictated by the surface tension (30 mN/m) of the NP solution.

Devices with pixel dimensions 300 × 900 µm were used to ensure sufficient loading of NPs in the device pixels because of the significantly larger sizes of NPs compared to organic dyes. Figure 4 shows transmittance spectra (a) and TEM image (b) of the nanoparticles. The spectra show increased transparency (at the nanoparticle absorption wavelength) as voltage is applied (20 V). This is a result of the Ag NP fluid system moving to the pixel grids and the water background fluid wetting the pixel and leading to greater device transparency. More importantly, significant attenuation (80%) of the peak intensity was achieved at 20 V. The reversible actuation of the device was also demonstrated, as shown by the spectra in Figure 4a. A nearly identical spectral profile was observed for the device before voltage application (0 V) and after voltage is removed (V OFF, green dashed curve), which suggests that there is minimal hysteresis in the device actuation.

#### 3.5.2. CdSe Nanotetrapods in EW Device

The actuation of NPs in EW devices was also demonstrated using CdSe NPs (tetrapods). These were specifically chosen because of the relatively narrow absorption peak (Appendix A) compared to other CdSe NPs. These were dispersed in limonene (90 mg/mL) and 90 uL of solution dosed in the device.

The concentration of NPs was dictated by the surface tension (27.7 mN/m) and minimizing aggregation of the NPs. Figure 5a shows the performance of a device before and after a field is applied. Interestingly, the device responded differently to voltage application compared to the previous devices. The spectrum at 20 V at 1 kHz (red curve) showed an overall increase in the opacity of the device at all wavelengths including the absorption peak maximum (508 nm). The results suggest that nanoparticles may be distributed at the interface of the polar and non-polar phases [16] due to incomplete dewetting of the pixels (residual nanoparticle/limonene in the pixels) as the voltage is applied. However, the NP peak was less defined (with the application of voltage) as observed with the previous devices. The less-defined NP peak suggests that the expected actuation-induced nanofluid dewetting from the device pixels still occurred, though not completely. In fact, the NP ‘peak trough’ at 508 nm is twice as deep at 0 V compared to 20 V. The removal of voltage (V OFF, green dashed curve) led to a complete recovery of the NP spectral profile confirming the reversible actuation of the device. Four cycles of device actuation (voltage ON and OFF) were performed and the device performance was identical with minimal to no hysteresis, as shown in Appendix A.

A different device was assembled using the same CdSe nanofluid used for the aforementioned device. The results in Appendix A show an identical spectral profile in the ON and OFF states as the previous device. This thus confirmed that the results were not dependent on the particular device used.

#### 3.5.3. CdSe Nanoplatelets in EW Device

Nanoplatelets (NPLs) having very narrow absorption peaks [35], as shown in Appendix A, were also investigated. Similarly, the concentration of nanomaterial (60 mg/mL) was chosen to maximize the colloidal stability of particles while achieving the desired surface tension (36 mN/m).

The transmittance spectra and TEM image of NPLs are shown in Figure 6a,b, respectively. The corresponding device is shown as an inset in Figure 6a. The spectra in Figure 6a show two exciton peaks at 432 nm and 461 nm for the NPLs prior to voltage application (0 V, black curve). The application of voltage (20 V, red curve) resulted in a >10% decrease in transmittance in the wavelength range from 480 nm to 750 nm and a <5% decrease in transmittance at wavelengths corresponding to the exciton peaks. Just as in the case of the tetrapods, the results suggest that nanoparticles may be distributed at the interface of the polar and non-polar phases [16] due to incomplete dewetting of the pixels (residual nanoparticle/limonene in the pixels) as the voltage is applied. Interestingly, the intensity of the exciton peak at 461 nm was significantly reduced while a broadening of the peak at 432 nm was observed. This was attributed to the aggregation of nanoparticles in the presence of water in the pixels (voltage-induced electrowetting of the pixels by water). The removal of voltage (V OFF) resulted in the recovery of the spectral properties of the NPLs as shown by the green dotted curve in Figure 6a.

The difference in the switchable optical properties of the anisotropic nanoparticles (tetrapods and nanoplatelets) compared to the spherical Ag nanoparticles can also be attributed to the tendency of anisotropic nanoparticles to self-assemble. Tetrapods have been shown to form porous networks [36] due to the linking of the ‘tetrapod arms’. NPLs also have a greater tendency to self-assemble because of their 2D morphology [35]. Regardless, this work has demonstrated switchable optical properties for nanomaterials with different sizes and morphologies. The study presented can thus be extended to a variety of nanomaterials having desired optical properties. There is ongoing work on evaluating the effect of different nanoparticle types and surface chemistries on the switchable optical properties.

### 3.6. Electro-Optic Response Times for Representative Dye and Nanoparticle Devices

The best-performing dye (BODIPY TMS) and nanoparticle (Ag nanoparticle) devices were further characterized to evaluate the switching times. Figure 7 shows the response times for dye and nanoparticle devices. Both devices show millisecond-range response times similar to those of previously reported electrowetting devices [22]. Applying an AC field induced a switching response from dark to transparent states with a response of 2–3 ms, but removing the applied AC field caused a slower relaxation from transparent to the initial dark state.

Interestingly, the nanoparticle device showed a significantly lower relaxation time (10 ms) compared to the dye device (80 ms). We believe that the nanoparticle surface ligand (oleylamine) may be functioning as a surfactant and influencing the interfacial tension of the nanofluid in the biphasic system. The interfacial tension primarily drives the relaxation of the fluids when the voltage is turned off, as indicated in Section 3.2. However, the effect of the nanoparticle ligand on the electro-optic response of the device is beyond the scope of this work. However, future work will be carried out to better understand this phenomenon.

## 4. Conclusions

The optical properties of BODIPY dyes and nanoparticles were extensively explored in electrowetting devices of different pixel dimensions. Switchable optical properties of dyes and nanoparticles were achieved at relatively low voltages. Reversible device actuation was achieved in dyes (dye 3) dosed in 200 × 600 µm devices. In addition, the structure of the dyes and device pixel dimensions were found to be important for realizing robust device performance. Devices dosed with representative plasmonic (Ag) and semiconductor (CdSe) nanoparticles were also successfully actuated. To the best of our knowledge, changing optical properties of nanoparticles in electrowetting optical shutter devices were demonstrated for the first time in this work. Nanoparticle-based devices showed reversible actuation with minimal hysteresis, as well as faster relaxation times. This work thus sheds more light on the potential use of a variety of dyes and nanoparticles for switchable optical applications.

## Data Availability

All relevant data are included in the main manuscript.

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
