# Peer review of "Switchable Optical Properties of Dyes and Nanoparticles in Electrowetting Devices"

_nanomaterials, 2024, doi:10.3390/nano14020142_

Round 1
Reviewer 1 Report
Comments and Suggestions for Authors
The article presents a study on optical switches using an electric wetting device, organic dyes, and nanoparticle solutions. It is a very promising research direction, and the authors conducted scientific experiments and analyzed the results. However, there are still some areas that need additional data to increase the scientific value of this study.
1. The article provides a detailed demonstration of the device's microstructure in Figure 1. Similar microstructure comparisons should be made for all functional devices to further demonstrate their value in optical switching, as this study is based on the optical switch research of the electric wetting pixel opening rate.
2. The article should present relevant data on the durability of the devices, such as whether the organic dyes used in the article have optical stability issues. It should also discuss whether nanoparticles may cause electrical leakage or other related problems after repeated switching of the devices.
3. The article should include a discussion of the mechanism of optical switches, combined with experimental results. At present, the article seems more like a technical report and lacks theoretical descriptions and discussions. For example, is the opening rate of the device the most important factor determining its performance?
4. During the experimental testing process, blank control test data should be added to eliminate optical attenuation caused by the glass devices themselves.
Comments on the Quality of English LanguagePlease check the whole paper to correct the wrong spelling of the words.
Reviewer 2 Report
Comments and Suggestions for Authors
In this work, Tongha et al. demonstrated switchable optical properties of dyes and NPs in liquid-based electrowetting on dielectric (EWOD) devices. Moreover, low voltage actuation of boron dipyromethene (BODIPY) dyes and NPs (Ag and CdSe) was demonstrated without degradation of the light-absorbing materials. Similarly, reversible and low voltage (≤ 20 V) switching of dye optical properties was also observed as a function of device pixel dimensions (300 x 900, 200 x 600, and 150 x 450 μ m). Furthermore, low voltage and reversible switching were also demonstrated for plasmonic and semiconductor NPs. It was observed that NP-based devices showed minimal hysteresis as well as faster relaxation times. The work is interesting and the obtained results are reasonable. Hence, I recommend this work for publication after minor revision. The authors must address the following concerns properly:
1) Please revise the abstract section especially lines 24-25.
2) Please include the different types of low-voltage EWOD devices and wetting phenomena in the introduction. Some important articles have been missed by the authors such as https://www.sciencedirect.com/science/article/abs/pii/S0924424715302326; https://pubs.rsc.org/en/content/articlelanding/2018/ta/c8ta08325h/unauth
3) Please provide the histogram for the average particle size distribution for different NPs.
4) Please provide the complete operating conditions of the instruments used for the characterization and EWOD measurements.
5) More discussion on switching optical properties/light attenuation is required from the nanoscale-sized particles.
6) What about using polar or nonpolar liquids? Did you try high-surface tension liquids?
Reviewer 3 Report
Comments and Suggestions for Authors
The manuscript by Tohgha et al. studied the electro-wetting behaviour of water/oil/dye, water/nanoparticles.
Different wetting effects via changing the dye and adding distinct nanoparticles are showed.
Relatively low hysteresis is observed for the case of the nanoparticle solution.
However, some parameters and content are not well described.
I would recommend to publish this work if the following concerns are addressed.
1. The parameters are somehow not clear. Table 1 shows the surface tension of difference dyes. I think this is the interfacial tension of dye-air. My understanding is that in experiments, the surrounding is water and the oil stays inside the water. If this is the case, the relevant parameter is the interfacial tension of water-oil/dye. I am wondering if the authors can measure or estimate the relevant interfacial tension of water-oil/dye. This question is also applied to the nanoparticle solution.
2. The current paper is discussing wetting effect. Within this scope, the contact angle, at least the equilibrium Young's contact angle should be given either V=ON or V=OFF. For example, I am wondering what is the contact angle of water on the ITO substrate, what is the contact angle of oil on the ITO substrate, what is the contact angle of water/oil on the pixel grids. It is not necessary to show the contact angle for V=ON; at least, the contact angle for V=OFF should be measured.
In addition, what are the contact angles when nanoparticle solutions are considered? I think the contact angle of a solution can be measured when it is dilute, similar to a coffee solution where nanoparticles are present.
In a more physical way, I am wondering if the particles considered in the present work are more favorite by the pixel grid or other areas of the substrate; whether it is affected by the geometric shape of the particles?
3. The hysteresis for different dyes is discussed in Figure 2.
I believe the contact angle hysteresis is closely related to the surface tension and device pixel dimensions,
as also claimed by the authors referring to Ref. [21]. I understand that the current work considers
a patterned substrate, on which the contact angle hysteresis has been comprehensively discussed in literature
such as, Wetting Effect on Patterned Substrate, Adv. Mater. 35, 2210745 (2023); Contact angle hysteresis, Current Opinion in Colloid & Interface Science 59 (2022): 101574; although the voltage is not applied.
But this is the same as the hysteresis for Fig. S1, Fig. S2, Fig. S3, section 3.2.
In addition, I am wondering what is the dimension of the pixel grid, which is most significant for the contact angle hysteresis.
It is confusion whether 300x900 um is the pixel grid or the size of the device.
4. In general, most of the experimental figures for the device lack a scale bar, such as Fig. 3b, the figure under Table 1, etc.
5. Minor grammars, page 5, line 226-227, The surface tension of the dyes...are shown should be The surface tension of the dyes...is shown.
A line break should be placed after line 115.
cf. comment 5.
Reviewer 4 Report
Comments and Suggestions for Authors
This manuscript demonstrates the switchable optical properties of dyes and nanoparticles in electrowetting on dielectric (EWOD) devices. It utilizes narrow band-absorbing dyes and nanoparticles for spectral-filtering applications targeting specific wavelengths. The study investigates boron dipyromethene (BODIPY) dyes and nanoparticles (Ag and CdSe) with low voltage actuation, examining their potential for light-attenuation applications​​. This device shows significant progress in the field of optics for EWOD devices, particularly in the application to plasmonic and semiconductor nanoparticles, exhibiting minimal hysteresis and faster relaxation times. The research content is interesting and with great implications.
Strengths
1. The research addresses a significant gap in light-attenuation applications, exploring the change in optical properties of materials in EWOD devices, an area with limited studies but great potential​​.
2. The materials and methods section are comprehensive, detailing the synthesis and properties of the dyes and nanoparticles used in the study​​.
3. Substantive Findings: The study effectively demonstrates switchable optical properties of both dyes and nanoparticles, with distinct behaviors based on their structures and device pixel dimensions. This provides a deeper understanding of the interplay between material properties and device performance​​.
Comments:
1. The authors can address how electrowetting will affect/drive the BODIPY dyes in the device.
2. Other than Transmittance/Absorption spectra, can you provide additional evaluation metrics to understand how well the electrowetting can be applied to drive the dyes/NPs?
3. The authors can elaborate the possible effect of nanoparticle ligand on the electro-optic response of the device in the Discussion session.
4. Please provide the important properties of dyes/NPs, related to electrowetting actuation.
5. What are the limitations to further improve the transmittance of these dyes under actuations?
6. The authors can provide additional table to benchmark the optical performance of these dyes/NPs, along with a few other dyes/NP commonly used in this field.
Round 2
Reviewer 1 Report
Comments and Suggestions for Authors
Thanks for the revised manuscript. The last few comments:
1. I would like to see the pixel change image under the microscope (voltage on and off) when the authors apply their experiments on their device. This will make sure the how the pixle worked.
2. Do the authors check how many times the device could swith on and off without any defects happened?
I am satisfied with the rest answer.
Author Response
We appreciate the valuable comments from the reviewer. Please see the attachment.

Reviewer 2 Report
Comments and Suggestions for Authors
The revised manuscript is now in better shape to be accepted for publication.
Author Response
We appreciate the valuable comments from the reviewer.